# Virtual Monochromatic Images from Dual-Energy Computed Tomography Do Not Improve the Detection of Synovitis in Hand Arthritis

**DOI:** 10.3390/diagnostics12081891

**Published:** 2022-08-04

**Authors:** Sevtap Tugce Ulas, Katharina Ziegeler, Sophia-Theresa Richter, Sarah Ohrndorf, Fabian Proft, Denis Poddubnyy, Torsten Diekhoff

**Affiliations:** 1Department of Radiology, Charité—Universitätsmedizin Berlin, Campus Mitte, Humboldt-Universität zu Berlin, Freie Universität Berlin, 10117 Berlin, Germany; 2Department of Rheumatology, Charité—Universitätsmedizin Berlin, Campus Mitte, Humboldt-Universität zu Berlin, Freie Universität Berlin, 10117 Berlin, Germany; 3Department of Rheumatology, Charité—Universitätsmedizin Berlin, Campus Benjamin Franklin, Humboldt-Universität zu Berlin, Freie Universität Berlin, 10117 Berlin, Germany

**Keywords:** dual-energy CT, ultrasound, virtual monochromatic imaging, arthritis, rheumatoid, synovitis

## Abstract

The objective of this study was to investigate subtraction images from different polychromatic and virtual monochromatic reconstructions of dual-energy computed tomography (CT) for the detection of inflammation (synovitis/tenosynovitis or peritendonitis) in patients with hand arthritis. In this IRB-approved prospective study, 35 patients with acute hand arthritis underwent contrast-enhanced dual-energy CT and musculoskeletal ultrasound (MSUS) of the clinically dominant hand. CT subtractions (CT-S) were calculated from 80 and 135 kVp source data and monochromatic 50 and 70 keV images. CT-S and MSUS were scored for synovitis and tenosynovitis/peritendonitis. Specificity, sensitivity and diagnostic accuracy were assessed by using MSUS as a reference. Parameters of objective image quality were measured. Thirty-three patients were analyzed. MSUS was positive for synovitis and/or tenosynovitis/peritendonitis in 28 patients. The 70 keV images had the highest diagnostic accuracy, with 88% (vs. 50 keV, 82%; 80 kVp, 85%; and 135 kVp, 82%), and superior sensitivity, with 96% (vs. 50 keV: 86%, 80 kVp: 93% and 135 kVp: 79%). The 80 kVp images showed the highest signal- and contrast-to-noise ratio, while the 50 keV images provided the lowest image quality. While all subtraction methods of contrast-enhanced dual-energy CT proved to be able to detect inflammation with sufficient diagnostic accuracy, virtual monochromatic images with low keV showed no significant improvement over conventional subtraction techniques and lead to a loss of image quality.

## 1. Introduction

A multitude of technical advances in the field of dual-energy computed tomography (DECT) in recent years have led to a wider spread in routine clinical practice [1,2,3]. Material specific attenuations in low- and high-energy spectra [4] allow for a relatively specific characterization of the material composition within a certain object [5]. The technique is most commonly used for the detection of uric acid crystal depositions in gout arthropathy in the musculoskeletal system [2].

In addition, virtual monochromatic images (VMIs) provide several advantages in the visualization of iodine contrast and reduction of metal implant artefacts [6]. VMI allows for the calculation of image data of different energy levels from a single CT scan. Low-energy VMIs simulate a low photon energy without the disadvantages of a broad spectrum, thus leading to an increase in the contrast between structures, for example, iodine contrast agent and soft-tissues.

The use of computed tomography (CT) in arthritis imaging has gained increasing interest in recent years [7,8,9]. Its high resolution enables precise and more sensitive detection of small erosions than in conventional radiography and MRI [10]. Especially in patients with early arthritis, sensitive imaging plays a crucial role in optimizing treatment and thus modifying the further disease course. This supplementary diagnostic information and the exclusion of differential diagnoses by imaging can help justify the initiation of anti-inflammatory treatment, which can be expensive.

So far, the detection of active inflammation by using CT has not played a major role. However, further developments in image postprocessing, such as CT subtractions (CT-Ss) from pre- and post-contrast images, allowed for the detection of synovitis and tenosynovitis [8]. In a recent study, the advantages of DECT and reconstruction of subtractions were presented in a cohort of patients with psoriatic arthritis [11]. While CT-S and DECT have shown to be more specific but less sensitive than MRI [12]—thus performing non-inferior for arthritis imaging—further means of improving the detection of iodine uptake in synovitis might compensate the shortfall in sensitivity. One way of achieving this might be the combination of two evaluated techniques, namely CT-S and DECT, by using virtual monochromatic images from DECT datasets for subtractions.

The aim of our study was to analyze the added value of monochromatic images for CT-S to detect synovitis and tenosynovitis/peritendonitis in hand arthritis. Standard-of-care musculoskeletal ultrasound (MSUS) was used as standard of reference.

## 2. Materials and Methods

The local ethic committee (EA4_005_18) and the Federal Office for Radiation Protection (Z5-22462/2-2019-039) approved the study. Written informed consent was obtained from all participants.

### 2.1. Subjects

In our prospective single-center study, we consecutively enrolled 35 patients with acute arthritis of the hand and the suspected or proven diagnosis of rheumatoid arthritis (RA) between October 2018 and October 2019. Exclusion criteria were patients with contraindications for CT contrast agent (hyperthyroidism, reduced kidney function (GFR < 60 mL/min and known allergic reactions to contrast agent). Only patients over the age of 18 were included. The final diagnosis was made by the treating rheumatologist of the local rheumatology department based on the clinical and laboratory findings and the results of the imaging.

### 2.2. DECT and MSUS Imaging Procedures

All patients underwent DECT, both before and 3 min after injection of body weight adapted contrast agent with 1 mL/kg Ultravist 370 (Bayer) and 30 mL isotonic saline with a flow of 3 mL/s. The scans were performed on a 320-row single-source CT scanner (Canon Aquilion ONE Vision, Canon Medical Systems, Otawara, Japan) and included a scanogram and a sequential volume acquisition with 16 cm *z*-axis coverage without table movement. Two energy datasets (135 and 80 kVp; 150 mAs) were measured with a rotation time of 0.275 s and a change-over time of 0.5 s, resulting in a total scan time of 1.05 s. Conventional 80 and 135 kVp images, as well as material maps, were primarily reconstructed with a thickness of 0.5 mm in a medium soft-tissue kernel and subject to further postprocessing.

A senior radiologist (T.D.) or senior rheumatologist (S.O.) with 10 years of experience in MSUS performed the MSUS. A high-frequency 24 MHz linear array transducer (Aplio 500, Canon Medical Systems) was used. The examination was performed according to the guidelines of the European Society of Musculoskeletal Radiology (ESSR). Inflammatory changes such as synovitis and tenosynovitis/peritendonitis were visualized by using the Power Doppler signal. For better comparability with the scoring results from CT, the grayscale and power Doppler images were graded by adapting a Rheumatoid Arthritis Magnetic Resonance Imaging Score (RAMRIS)-derived scoring system which included the assessment of the metacarpophalangeal and proximal interphalangeal joints II–V, as well as the radioulnar, radiocarpal, and intercarpal joints. Furthermore, the flexor and extensor tendons were separately assessed for tenosynovitis and peritendonitis. According to the RAMRIS synovitis score, ratings between 0 and 3 were used.

### 2.3. VMI and Subtraction

VMI with energy levels at 50 and 70 keV were generated from the raw datasets by using the CT console (dual-energy image view and dual-energy raw data analysis, version 6, Canon Medical Systems). Both pre- and post-contrast scans were subject to this reconstruction. VMI and conventional CT images were then postprocessed by using a subtraction software (SureSubtraction Ortho Version 5, Canon Medical Systems) to obtain a bone-free color-coded multiplanar image dataset with a slice thickness of 3.0 mm, while adjusting the windowing to the expected noise-level. Examples of grayscale subtraction images are presented in Figure 1.

### 2.4. Image Reading

All images were pseudonymized for name, age, sex, and the energy level.

Two independent readers (K.Z. with 7 years and S.T.U. with 3 years of experience in musculoskeletal imaging), who were blinded to identifying information, clinical data, and other imaging findings, scored the CTs for synovitis, tenosynovitis, and peritendonitis according to the RAMRIS criteria, defined by the Outcome Measurement in Rheumatology (OMERACT) MRI group [13]. Disagreement was solved in a consensus session. Disagreement was defined in regions of scoring results RAMRIS = 0 and RAMRIS = 1 and scoring differences of ≥2. MSUS images were scored by the examiners themselves, according to the RAMRIS criteria.

### 2.5. Image Quality

For quantitative evaluation of the image quality, one radiologist (S.T.U.) performed a region of interest (ROI) analysis and measured the mean and standard deviation of CT number in Hounsfield Units (HU) with a dedicated software (dual-energy image view, version 6, Canon Medical Systems). The ROI was placed in an inflamed (Syn) and another in a non-inflamed joint (NSyn), according to the results of the RAMRIS image reading. For reference, additional ROIs were placed in the thenar muscle, as well as in the radial artery. The location of the ROI was automatically transferred to the entire (monochromatic) energy levels of the CT-S. A size of 4.9 × 4.9 mm^2^ was chosen for all ROI measurements.

The image quality was assessed by using the signal-to-noise ratio (SNR) and contrast-to-noise ratio (CNR) as follows:(1)$$\mathrm{SNR}=\frac{\mathrm{HU}\;\mathrm{artery}}{\mathrm{standard}\;\mathrm{deviation}\;\mathrm{muscle}}$$
(2)$$\mathrm{CNR}=\frac{\mathrm{HU}\;\mathrm{Syn}-\mathrm{HU}\;\mathrm{NSyn}}{\mathrm{standard}\;\mathrm{deviation}\;\mathrm{muscle}}$$

### 2.6. Radiation Exposure

The radiation exposure (estimated effective dose) of DECT examinations was calculated by using the overall dose-length product (DLP) and a conversion coefficient of 0.0008 [mSv × mGy ^–1^ × cm ^–1^].

### 2.7. Statistical Analysis

The statistical analysis was performed by using GraphPad Prism (Version 7 for MacOS, GraphPad Software, La Jolla, CA, USA). The scoring results of both readers were combined by calculating the mean scores. For the comparison of the scoring results, they were dichotomized and summarized. Synovitis, tenosynovitis, and peritendonitis detected in MSUS were used as reference. Patients with one or more regions with presence of inflammation were categorized as synovitis-positive. Using the one-tailed McNemar test, the number of synovitis-positive patients in virtual monochromatic and conventional CT-S was compared to MSUS to test for non-inferiority. For further statistical purposes, contingency tables were created for each virtual monochromatic and conventional CT-S separately. Sensitivity, specificity, diagnostic accuracy, and positive and negative predictive value were calculated. Sum scores were calculated to test for significant correlations, using a Pearson test. The inter-rater reliability was calculated by using the Intraclass Correlation Coefficient (ICC). The image quality of the different energy level CT-S was compared by using the Wilcoxon matched-pairs signed rank test. A *p* value <0.05 was considered significant.

## 3. Results

### 3.1. Subjects

Two patients had to be excluded from further analysis due to technical problems with the reconstruction of the VMI. The flowchart of study inclusion is presented in Figure 2.

Thus, the study included 33 patients (21 women and 12 men). In Table 1, the patients’ characteristics are summarized. Fifteen patients were treatment-naïve, seven received conventional synthetic (cs) DMARDs, six received corticosteroid solely, and one received biological (b)DMARD. Three patients were treated with a combination of corticosteroids and csDMARD, and one patient with a combination of csDMARD and bDMARD.

### 3.2. Image Reading and Statistical Analysis

Examples of subtraction images are presented in Figure 3 and Figure 4.

In total, 28 patients were positive for synovitis and/or tenosynovitis/peritendonitis on MSUS (mean sum score 6.91 ± 7.76), 29 patients on CT-S with 80 kVp (mean sum score 6.38 ± 7.63), 22 patients on CT-S with 135 kVp (mean sum score 4.45 ± 6.40), 30 patients on CT-S with 70 keV (mean sum score 6.62 ± 7.26), and 26 patients on CT-S with 50 keV (mean sum score 4.77 ± 6.22). The results of the contingency table analyses are presented in Table 2 and in Appendix A. On the patient-level, CT-S with 80 kVp yielded a sensitivity of 93% (95% CI 77% to 99%), 70 keV of 96% (95% CI 83% to 99%), and 50 keV of 86% (95% CI: 69% to 94%). The lowest sensitivity was calculated for 135 kVp (79% (95% CI: 60% to 90%)). The highest diagnostic accuracy was shown for 70 keV (88%), followed by 80 kVp (85%). Moreover, 50 keV and 135 kVp achieved a diagnostic accuracy of 82%.

In total, 693 regions (joints and tendons) were included in the analysis. MSUS detected inflammation in 155 regions. Moreover, 80 kVp was positive in 144 regions, 70 keV in 150 regions, 50 keV in 114 regions, and 135 kVp in 105 regions. Using the McNemar test, we found that the 70 keV (*p* = 0.31), 50 keV (*p* = 0.34), and 80 kVp (*p* = 0.50) images showed non-inferiority compared to MSUS, but not 135 kVp (*p* = 0.02). The highest sum score correlation was presented for 80 kVp (r = 0.912, *p* < 0.0001), followed by 50 keV (r = 0.901, *p* < 0.0001), 70 keV (r = 0.884, *p* < 0.0001), and 135 kVp (0.873, *p* < 0.0001).

The assessment of the ICC showed no significant difference between virtual monochromatic and conventional CT-S (ICC 70 keV = 0.73, ICC 50 keV = 0.76, ICC 80 kVp = 0.77, and ICC 135 kVp = 0.77).

### 3.3. Image Quality

In Figure 5, the results of the image quality analysis are presented. The highest SNR and CNR were shown for CT-S reconstructions with 80 kVp. Reconstructions with monochromatic energy levels showed significantly lower SNR and CNR, especially for 50 keV. A higher energy level of 135 kVp provided an improved SNR compared to the monochromatic reconstruction, but it also led to a lower CNR at the same time. Therefore, the difference of the CNR between reconstructions with 70 keV and 135 kVp was not statistically significant. A lower energy level leads to an increase in image noise.

### 3.4. Radiation Exposure

The total DLP was 93.2 mGy × cm, with an estimated effective dose of 0.075 mSv.

## 4. Discussion

To the best of our knowledge, this is the first study to investigate the subtraction of VMIs for the detection of synovitis and tenosynovitis/peritendonitis in a cohort of patients with hand arthritis. Conventional CT-S and virtual monochromatic CT-S showed moderate sensitivity and high specificity when using MSUS as the standard of reference. However, virtual monochromatic CT-S did not achieve any additional advantages over conventional CT-S. Compared to MSUS, all reconstructions showed non-inferiority, except for 135 kVp. An almost-perfect correlation was shown for conventional and virtual monochromatic CT-S compared to MSUS.

Recently, a few studies have investigated CT for the detection of inflammatory joint disease and, especially, active soft-tissue inflammation by using contrast enhancement [8,11,14]. Compared to MSUS, which has become established imaging in early peripheral arthritis [15], CT has the decisive advantage of superior standardization and investigator-independent analysis. In addition, due to its high resolution, CT enables more reliable detection of structural changes such as erosions or soft-tissue calcifications [10,16]. For an example, several studies have shown that DECT is superior to MSUS in the detection and characterization of urate deposits in gouty arthritis [17,18]. In contrast to MSUS, studies showed that gout deposits cannot be reliably detected below a certain concentration in DECT [19,20]. CT allows for the assessment of joints and parts of joints that are otherwise hidden from the ultrasound probe, while not overcalling minor inflammation, which has been shown for MRI [12]. However, a shortfall in sensitivity impedes the widespread use of CT-S and similar techniques in clinical practice.

VMI is one of the reconstruction techniques intrinsic to DECT. The main purpose of VMI is to optimize imaging for routine diagnostics that is adapted to the requirements [21,22]. It is assumed that VMI with 70 keV corresponds most closely to conventional CT with 120 kVp [23]. In our study, 70 keV showed sensitivities and diagnostic accuracies comparable to those of CT-S at 80 kVp. Using lower-energy-level VMI leads to improved contrast, accompanied by an increase in image noise. This resulted in a lower SNR and CNR from using lower-energy-level VMI. Both effects combined did not lead to a net-gain of diagnostic accuracy for CT-S. The highest SNR and CNR were showed for conventional CT-S with 80 kVp. Higher energy levels with 135 kVp did not provide increased SNR and CNR.

A further reconstruction technique using DECT is to generate iodine maps according to the specific atomic numbers of iodine. In the study of Fukuda et al., the potential of DECT iodine maps was shown in regard to the detection of synovitis in patients with psoriatic arthritis [7,11]. In a further study by this group, DECT iodine maps were also successfully used in therapy monitoring [24] and in the anatomical analysis of psoriatic arthritis [25]. Nonetheless, a recent study found no advantage of DECT over CT-S for synovitis detection [12].

While the most suitable CT technique for arthritis imaging has yet to be established, our findings suggest that the combination of DECT and CT-S does not significantly improve the diagnostic performance. CT-S is a very robust technique that, in theory, is more sensitive to contrast-uptake than DECT iodine maps [26], but it might be more susceptible to motion during the scan time. DECT, on the other hand, does not fully subtract bone or calcium within the soft tissues, and artefacts in the vicinity of bone or skin might hamper the easy interpretation of images. However, other DECT reconstructions show an ever-increasing potential for clinical practice [27]. For example, DECT allows for the detection osteitis, thus providing additional information about bone marrow inflammation that has not been analyzed for CT-S [1,28]. This is important, as upcoming photon-counting detector technology is likely to provide spectral information for all CT-scans in the near future [29].

MSUS is a well-established imaging tool in routine clinical practice in the early detection of synovitis and is recommended by the current guidelines for early arthritis [30]. Therefore, we deliberately selected MSUS as our standard of reference. It enabled us to depict synovial hypertrophy with high sensitivity and specificity even without the application of contrast media. Using power Doppler, hyperemia in inflamed joints can be sufficiently detected [31]. The major disadvantage of MSUS is the dependence of the diagnostic accuracy on the experience of the examiner [32]. For better comparability, established US scoring systems were not used; instead, we used a score adapted from MRI. Due to its low specificity, we decided against the use of MRI for this analysis [30]. Our explorative study included only 33 patients. Nevertheless, we were able to receive significant results. In addition, the patient cohort was heterogeneous. However, we assume that this fact corresponds to the heterogeneous patient collective in routine clinical practice, and we focused on imaging findings (synovitis/tenosynovitis or peritendonitis) that were present in most of the included joint diseases.

## 5. Conclusions

In this study, we investigated the value of VMI in the detection of synovitis and tenosynovitis, using MSUS as standard of reference. We showed that VMI could visualize inflammatory changes with high diagnostic accuracy and sensitivity. However, VMI with lower energy levels does not have a major positive effect on the performance of CT-S.

## Figures and Tables

**Figure 1 diagnostics-12-01891-f001:**
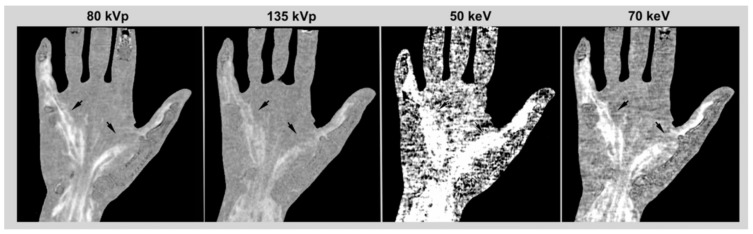
Examples of grayscale CT subtractions with conventional 80 and 135 kVp images and 50 and 70 keV monochromatic images. Images are presented with WL/WW of 0/300. While high-energy datasets (135 kVp) showed low noise and contrast, the contrast increases with lower energy and for monochromatic reconstructions.

**Figure 2 diagnostics-12-01891-f002:**
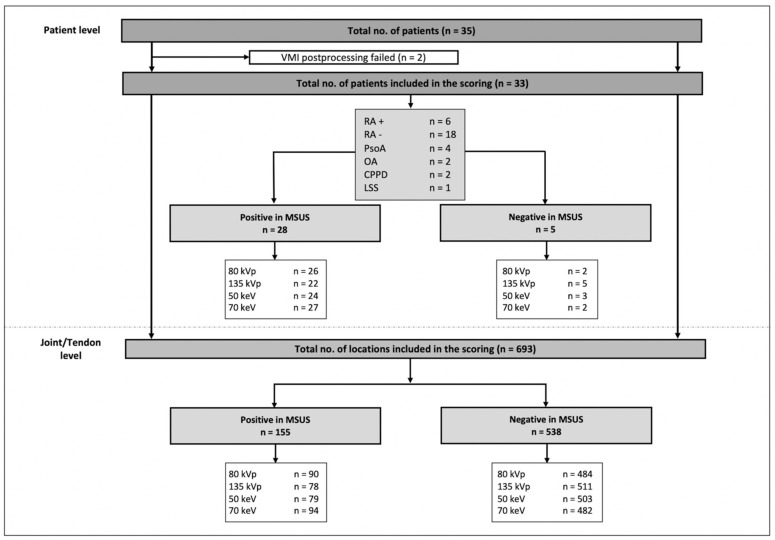
Flowchart of study inclusion and results of the RAMRIS scoring. VMI = virtual monochromatic imaging, RA+ = seropositive rheumatoid arthritis, RA− = seronegative rheumatoid arthritis, PsoA = psoriatic arthritis, OA = osteoarthritis, CPPD = calcium pyrophosphate deposition disease, LSS = limited systemic sclerosis, MSUS = musculoskeletal ultrasound.

**Figure 3 diagnostics-12-01891-f003:**
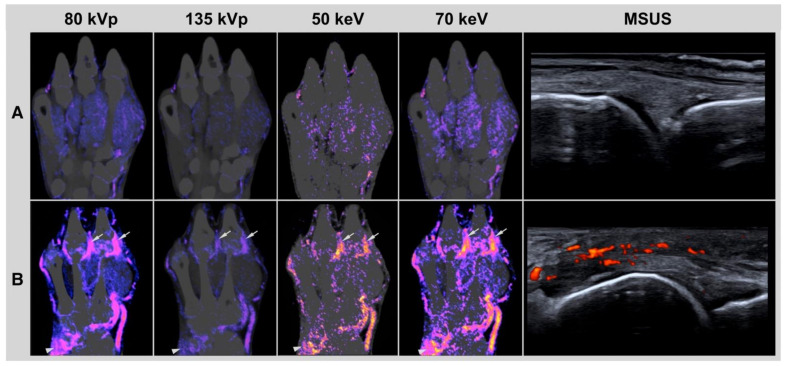
Imaging examples. Conventional CT subtraction for 80 and 135 kVp. Virtual monochromatic imaging CT subtraction for 70 and 50 keV. MSUS: musculoskeletal ultrasound of the metacarpophalangeal (MCP) joint II from the dorsal view. (**A**) Imaging examples of a 51-years old patient with seropositive rheumatoid arthritis. No inflammatory activity was detected. (**B**) Imaging examples of a 62-years old patient with treatment-naïve seronegative rheumatoid arthritis. Severe synovitis was detected at the metacarpophalangeal (MCP) joint III and II (arrows) and wrist (arrowhead). In 135 kVp, the synovitis in these regions was underestimated. MSUS of the MCP II joint showed synovitis and peritendonitis with severe power Doppler activity.

**Figure 4 diagnostics-12-01891-f004:**
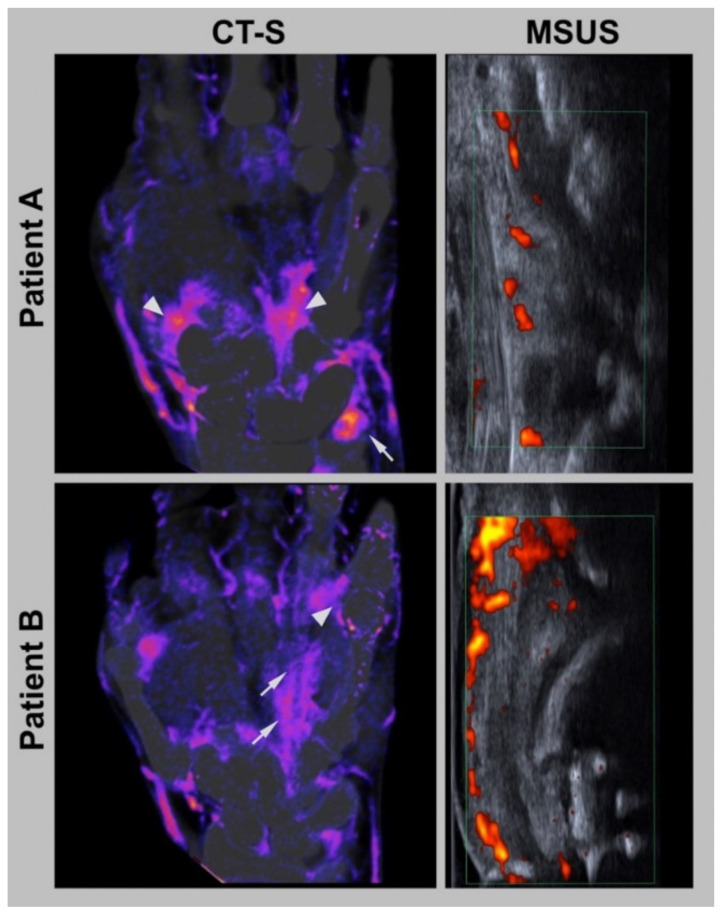
Patient examples. CT-S = conventional CT subtraction for 80 kVp, MSUS = musculoskeletal ultrasound of the wrist (Patient A) and of the metacarpophalangeal (MCP) joint V from the dorsal view (Patient B). Patient A: Imaging examples of a 75-year-old male patient with CPPD. CT-S and MSUS showed severe synovitis in the wrist (arrowhead) and on the processus styloideus ulnae (arrow). Patient B: Imaging examples of a 62-year-old man patient with seronegative rheumatoid arthritis. Severe synovitis was detected at the MCP V joint in CT-S and MSUS (arrows). Furthermore, CT-S showed severe tenosynovitis of the ring finger (arrowhead).

**Figure 5 diagnostics-12-01891-f005:**
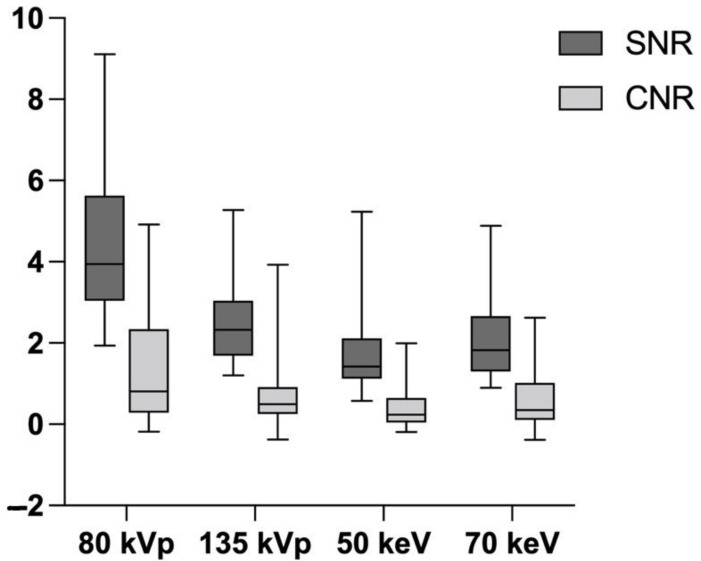
Quantitative assessment of image quality. SNR = signal-to-noise ratio, CNR = contrast-noise ratio. The highest SNR and CNR were shown for CT-S with 80 kVp. CT-S with monochromatic energy levels (50 and 70 keV) provided significantly lower CNR and SNR values, which resulted in lower image quality.

**Table 1 diagnostics-12-01891-t001:** Results of the descriptive analysis. ACPA = antibodies against citrullinated peptides, RF-IgM = rheumatoid factor IgM, y = years, *n* = number, SD = standard deviation.

Characteristics	
Number of patients (women/men)	33 (21/12)
Mean age (y) (SD; range)	55 (11.8; 23–75)
Mean symptom duration (y) (SD; median; range)	1.7 (3.8; 0; 0–16)
Mean CRP (mg/L) (SD; median; range)	18.3 (30.3; 3.4; 0.3–132.0)
ACPA-positive (>20 U/mL), n	6 (18.2%)
RF-IgM-positive (>14 U/mL), n	12 (36.4%)

**Table 2 diagnostics-12-01891-t002:** Results of the contingency table analysis. SE = sensitivity, SP = specificity, PPV = positive predictive value, NPV = negative predictive value, 95% CI = 95% confidence interval of the difference.

Patient Level	SE (95% CI)	SP (95% CI)	PPV (95% CI)	NPV (95% CI)
**80 kVp**	0.93 (0.77–0.99)	0.4 (0.07–0.77)	0.90 (0.74–0.96)	0.5 (0.09–0.91)
**135 kVp**	0.79 (0.60–0.90)	1.0 (0.57–1.00)	1.0 (0.85–1.0)	0.45 (0.21–0.72)
**50 keV**	0.86 (0.69–0.94)	0.6 (0.23–0.93)	0.92 (0.76–0.99)	0.43 (0.16–0.75)
**70 keV**	0.96 (0.83–0.99)	0.4 (0.07–0.77)	0.9 (0.74–0.97)	0.67 (0.12–0.98)
**Joint/Tendon Level**				
**80 kVp**	0.58 (0.50–0.66)	0.90 (0.87–0.92)	0.63 (0.54–0.70)	0.88 (0.85–0.90)
**135 kVp**	0.50 (0.43–0.58)	0.95 (0.93–0.97)	0.74 (0.65–0.82)	0.87 (0.84–0.89)
**50 keV**	0.51 (0.43–0.59)	0.93 (0.91–0.95)	0.69 (0.60–0.77)	0.87 (0.84–0.89)
**70 keV**	0.61 (0.53–0.68)	0.90 (0.87–0.92)	0.63 (0.55–0.70)	0.89 (0.86–0.91)
**Joint Level**				
**80 kVp**	0.56 (0.47–0.64)	0.89 (0.85–0.92)	0.71 (0.61–0.79)	0.81 (0.76–0.85)
**135 kVp**	0.47 (0.38–0.56)	0.95 (0.91–0.97)	0.81 (0.70–0.88)	0.79 (0.74–0.83)
**50 keV**	0.50 (0.42–0.59)	0.90 (0.85–0.93)	0.70 (0.60–0.79)	0.79 (0.74–0.84)
**70 keV**	0.62 (0.52–0.70)	0.86 (0.81–0.90)	0.67 (0.58–0.75)	0.82 (0.77–0.87)
**Tendon Level**				
**80 kVp**	0.66 (0.50–0.79)	0.91 (0.87–0.94)	0.48 (0.35–0.61)	0.95 (0.92–0.97)
**135 kVp**	0.61 (0.45–0.74)	0.95 (0.92–0.97)	0.62 (0.46–0.76)	0.95 (0.92–0.97)
**50 keV**	0.53 (0.37–0.68)	0.97 (0.94–0.98)	0.67 (0.49–0.81)	0.94 (0.91–0.96)
**70 keV**	0.58 (0.42–0.72)	0.93 (0.89–0.95)	0.51 (0.37–0.65)	0.94 (0.91–0.97)

## Data Availability

The data presented in this study are available upon request from the corresponding author.

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
