# Peer review of "Virtual Monochromatic Images from Dual-Energy Computed Tomography Do Not Improve the Detection of Synovitis in Hand Arthritis"

_diagnostics, 2022, doi:10.3390/diagnostics12081891_

Round 1

Reviewer 1 Report

Thank you for the opportunity to review this article.

The topic is very interesting. I think the implementation of new methods to aid diagnosis in patients with inflammatory joint diseases is crucial.

The study is well conducted. The article is well written. In particular, the methods are detailed and the statistical analyses are apparently adequately conducted. The discussion is also comprehensive.

I think the article is worthy of publication. I only have a few suggestions for the authors:

1) the introduction should be expanded and some points (especially regarding VMI) further simplified to facilitate the understanding of non-radiologist readers;

2) the inclusion of figures from another case would be welcome;

3) the conclusions deserve an independent paragraph.

Thank you.

Reviewer 2 Report

Possibility to visualize inflammatory changes in hand arthritis is of great importance for physicians included in treatment process. All relevant and recently published literature should be included in manuscript and discussed/compared to authors findings, for example Ogiwara S et al 2021. Additionally it would be interesting to discuss advantages/disadvantages of DECT vs ultrasound (Klauser AS et al, 2018).

Reviewer 3 Report

The paper analyzes the value of CT subtraction from different polychromatic and virtual monocromatic images in detection of synovitis, tenosynovitis and peritendonitis in hand arthritis in comparison to ultrasound. The subtraction images were able to detect inflammation with good diagnostic accuracy, however virtual monochromatic images showed no significant improvement.

The paper is interesting and sound. 

I have only few comments for the authors:

- Page 3, line 120. RAMRIS has been defined in line 96.

- Page 3. Please state the ultrasound machine used. 

- Page 3. Please clarify why for the US scoring was used and adapted version of the RAMRIS, instead of other validated US scoring methods (eg: GLOESS).
